# Effects of Different Marinades and Types of Grills on Polycyclic Aromatic Hydrocarbon Content in Grilled Chicken Breast Tenderloins

**DOI:** 10.3390/foods13213378

**Published:** 2024-10-24

**Authors:** Marta Ciecierska, Urszula Komorowska

**Affiliations:** Department of Food Technology and Evaluation, Institute of Food Sciences, Warsaw University of Life Sciences, Nowoursynowska 159 Street, 02-787 Warsaw, Poland; ula.komorowska@o2.pl

**Keywords:** PAHs, chicken breast tenderloins, grill, marinades, QuEChERS, HPLC–FLD/DAD

## Abstract

Grilling has become a widespread method of thermal food processing. However, food prepared in this way may be a source of carcinogenic organic compounds, such as polycyclic aromatic hydrocarbons (PAHs). The present study aimed to evaluate the impact of different marinades and grilling tools on PAH contamination of chicken breast tenderloins. Together with the determination of PAHs carried out using the QuEChERS–HPLC–FLD/DAD method, the meat’s weight loss after the thermal process and the color of raw and grilled samples were analyzed. Statistically, the highest levels of PAH contamination were found in samples prepared on a charcoal grill without a tray, whereas the lowest were seen using the ceramic contact grill. Meat marination showed that universal and chicken marinades can be barriers against PAHs. Following requirements set in Commission Regulation (EU) No. 915/2023, none of the analyzed samples exceeded the maximum allowable level for B[a]P (5.0 µg/kg) and the sum of four marker-heavy PAHs (30.0 µg/kg). Thus, preparing meat before the thermal process, including marinades rich in phenolic compounds, and selecting a grilling method with appropriate grilling tools can ensure food safety and effectively reduce PAH contamination in grilled poultry meat.

## 1. Introduction

Polycyclic aromatic hydrocarbons (PAHs), or polyarenes, represent a diverse class of organic chemical compounds containing at least two fused aromatic rings of carbon and hydrogen atoms. The adverse effects of PAHs on the human body are primarily determined by their structure. Polycyclic aromatic hydrocarbons with 2–3 or 2–4 benzene rings, depending on categorization, are classified as light PAHs, and PAHs with at least 4 or 5 rings in their structure are classified as heavy PAHs [1,2].

Various organizations have engaged in research into the adverse effects of PAHs on human health. In 1976, the United States Environmental Protection Agency (US EPA) established a list of 16 PAHs as indicator compounds. This list included both highly carcinogenic PAHs and those with low toxicity to humans [3]. Nevertheless, compared to the 16 PAHs from the US EPA list, in December 2002, the European Union Scientific Committee on Food (SCF) identified 15 heavy PAHs as much more genotoxic, mutagenic, and carcinogenic and recommended them for monitoring in food. They were as follows: benzo[a]pyrene (B[a]P), benzo[a]anthracene (B[a]A), benzo[b]fluoranthene (B[b]F), benzo[j]fluoranthene (B[j]F), benzo[k]fluoranthene (B[k]F), benzo[g,h,i]perylene (B[ghi]P), chrysene (Chr), cyclopenta[c,d]pyrene (C[cd]P), dibenzo[a,h]anthracene (D[ah]A), dibenzo[a,l]pyrene (D[al]P), dibenzo[a,e]pyrene (D[ae]P), dibenzo[a,h]pyrene (D[ah]P), dibenzo[a,i]pyrene (D[ai]P), indeno[cd]pyrene (I[cd]P), and 5–methylchrysene (5-MChr) [4]. In 2005, the Joint FAO/WHO Expert Committee on Food Additives (JECFA) classified 13 of them as carcinogenic and mutagenic compounds and recommended further analysis of heavy polyarenes. The International Agency for Research on Cancer (IARC) classified polyarenes into four groups of carcinogenicity: group 1—human carcinogens, group 2A—probable carcinogens, group 2B—possible carcinogens, group 3—not classifiable as human carcinogens, and group 4—probably not carcinogenic to humans [5]. The reference in this classification was benzo[a]pyrene, the most common and studied PAH compound, with a toxicity value of 1. Several years ago, benzo[a]pyrene was considered to be a good marker for the occurrence of PAHs in food products. However, the 2005 European Commission Recommendation and the EFSA Panel on Contaminants in the Food Chain (CONTAM) revealed that B[a]P is not an appropriate indicator of other PAHs in food products [5,6,7]. Subsequently, the sum of four heavy PAHs (benzo[a]pyrene, chrysene, benzo[b]fluoranthene, and benzo[a]anthracene) has been established as the best marker of PAHs’ occurrence in food [8].

PAHs are found in different environmental compartments, such as water, air, or soil [9,10,11]. They are formed via various anthropogenic activities, especially combustion processes, and during forest fires and volcanic eruptions. Food is contaminated due to environmental deposition and specific thermal treatment processes and is the primary source of human exposure to PAHs [8,10,12,13]. According to some studies, it has been reported that there are three main potential mechanisms of PAH formation: pyrolysis of organic substances (fats, proteins, carbohydrates) at high temperatures, incomplete combustion of fuels, and dripping of fat onto open flames [14].

Grilling is one of the most common meat processing methods, especially in the summer. This cooking technique allows consumers to quickly prepare food with a unique taste, color, and aroma. However, many carcinogenic substances, including polycyclic aromatic hydrocarbons, are formed during this thermal process. The primary factors influencing PAH contamination in grilled foodstuffs are the type of meat, its fat content, the processing time and temperature, the method of cooking, and the distance from the heating source [8,15]. In the direct method of grilling, the dripping fat from the meat onto the hot coals generates carcinogenic substances such as PAHs, which are deposited on the surface of the food, thus contributing to high concentrations of polyarenes [9]. To minimize their formation, an indirect method of grilling is often used. Shortening the grilling time or avoiding direct contact with the flames using an aluminum tray can significantly minimize the formation of PAHs. Instead of charcoal grills, other types, such as electric and gas grills, are increasingly used to reduce PAH contamination [15,16].

Intensive research is ongoing on preventing PAHs in food, especially processed foods. In addition to choosing the appropriate cooking technique, preparing the raw materials, including marinating, can significantly reduce PAH formation. Marinades with acids and phenolic compounds can minimize the PAH content in the final products. Furthermore, marinating meat improves its texture and color and prolongs its shelf life [17,18,19].

Because of the high PAH contamination risk in grilled meat products and their regular consumption by consumers, especially in the summer season, maximum permissible levels for polyarenes, including the sum of four marker-heavy SCF PAHs and benzo[a]pyrene, were set in Commission Regulation (EU) No. 915/2023 for grilled meat products, at levels of 30 and 5 µg/kg, respectively [20].

In light of the above, examining PAH formation and occurrence levels in grilled meat products and developing treatments and methods to prevent and reduce PAH contamination in such products are essential. While many scientific publications address minimizing PAH levels in processed meats, they often focus on the list of 16 EPA PAHs. However, according to the SCF and EFSA opinions and recommendations mentioned above, researchers should prioritize the more toxic PAHs listed by the SCF. Therefore, the scope of this study was the examination of chicken breast tenderloins, the most frequently grilled type of poultry meat, prepared in different types of marinades commonly used in Poland, and subsequently grilled using various grilling tools. PAH determination was performed using one of the modern eco-friendly extraction methods, the QuEChERS method, and liquid chromatography with fluorescence and diode-array detectors (QuEChERS-HPLC–FLD/DAD). This work aimed to select which type of grilling tool and marination contributed to minimizing the PAH content in the analyzed products. In light of the aforementioned, PAH determination covered 19 polyarenes, comprising 15 heavy SCF PAHs with B[a]P and 4 marker-heavy PAHs, in addition to the 4 light EPA PAHs, which are typically predominant in profiles of PAH contamination. They were also analyzed, primarily for comparison with previous studies. These research data will contribute further insight into the effects of the grilling method and marinating on the levels of PAH contamination in grilled poultry meat, as well as proposing strategies for PAH reduction to minimize consumers’ dietary exposure to these toxic compounds.

## 2. Materials and Methods

### 2.1. Research Material, Its Preparation for Grilling, and Experimental Design

The materials investigated were raw chicken breast tenderloins purchased from a local market in Warsaw, Poland. Samples were cut into 100 g pieces and marinated in three previously prepared marinades: universal, chicken, and honey mustard marinades, all typically used in Poland for poultry marination. These marinades were chosen based on their popularity among consumers. The marinades were prepared according to the manufacturer’s instructions (the formulation of the marinades is given in Table 1).

The marinated samples were stored in glass food containers in a refrigerator at 4 ± 1 °C overnight. The unmarinated chicken breast tenderloins were treated identically, but no marinating was involved. Samples were prepared in triplicate, and the grilling process was carried out in 2 batches (*n* = 6, six samples of every kind of product were analyzed). The following day, before grilling, the chicken breast tenderloins were placed at room temperature until the internal temperature of the samples reached 15–20 °C (HANNA Instrument HI 98804 thermometer, Woonsocket, RI, USA).

The experimental design included marinating the meat in previously prepared marinades, heat treatment using different types of grills, assessment of weight loss after the grilling process, measurement of color parameters of raw and grilled products, sample preparation for PAH determination using the QuEChERS method, and qualitative and quantitative analysis of PAH content using a liquid chromatograph with fluorescence and diode-array detectors (HPLC–FLD/DAD).

### 2.2. Grilling Tools and Cooking Procedure

To assess whether the grilling method and the type of grill have an impact on the formation of PAHs, the following grilling tools were used: a charcoal grill without a tray (Weber Original Kettle E-4710, Weber-Stephen Deutschland GmbH, Ingelheim am Rhein, Germany), the same charcoal grill with an aluminum tray, an electric cast-iron contact grill with corrugated upper and bottom surfaces (Combi Grill, GR-1000 model, Optimum, Mińsk Mazowiecki, Poland), and an electric ceramic contact grill with a corrugated upper surface and smooth bottom surface (SpidoCook, XP010PR model, UNOX, Cadoneghe, Italy).

Firstly, the appliances were preheated for at least 10 min. During that time, the weights of the chicken breast tenderloin samples before cooking were recorded. Once the appliances were preheated, the meat samples were placed on the grill surface. The grilling parameters for the particular types of grills are given in Table 2.

Once reaching the preferred medium degree of meat doneness and the desired temperature inside the product’s geometric center (>80 °C), the samples were removed from the grilled surface, cooled to room temperature, reweighed, and packed into glass containers for subsequent PAH analysis.

### 2.3. Weight Loss

To determine the weight loss, the weights of the chicken breast tenderloin samples were recorded in triplicate before and after grilling. The equation for calculating the cooking loss in grilled samples is provided below:$$Weight\;loss=\frac{M-M_{g}}{M}\cdot 100\%$$

M—weight of raw chicken breast tenderloin sample;

M_g_—weight of grilled chicken breast tenderloin sample.

### 2.4. Color Measurement

CIE L*a*b* parameters were measured for raw and grilled samples using a Minolta Colorimeter (model: Chroma Meter CR–200; Konica Minolta Corp., Tokyo, Japan) calibrated with a white plate (L* − 97.83, a* − 0.45, b* + 1.88). Based on the lightness (L*), redness (a*), and yellowness (b*), the color differences between samples before and after grilling (ΔE) were calculated.
∆E = [(∆L)^2^ + (∆a)^2^ + (∆b)^2^]^0.5^

### 2.5. Chemicals and Materials

Acetonitrile (HPLC gradient grade), sodium chloride, and anhydrous magnesium sulfate (analytical purity > 99.0%) were obtained from Avantor Performance Materials Poland S.A. (Gliwice, Poland). The sorbents used in the QuEChERS method were primary secondary amine, silica gel modified with C18 groups, and graphitized carbon black (Sepra PSA Bulk Packing, Sepra C18–E Bulk Packing, and Sepra GCB Bulk Packing, respectively) and were supplied by Phenomenex (Warsaw, Poland). Standard mixtures of 15 PAHs from the SCF list (PAH–Mix 183, Dr. Ehrenstorfer) and 16 US EPA PAHs (PAH–Mix 9, Dr. Ehrenstorfer) were provided by Witko (Łódź, Poland). The 16 US EPA PAH standard mixture was utilized only for the determination of 4 light PAHs, i.e., phenanthrene (Phen), anthracene (Anthr), fluoranthene (F), and pyrene (Pyr). Deionized water was sourced from a Millipore Milli-Q water purification system. Polytetrafluoroethylene syringe filters (PTFE, 25 mm i.d., 1 µm pore size) and Falcon centrifuge tubes (PTFE) were both supplied by BioAnalytic (Gdańsk, Poland).

### 2.6. Determination of PAHs Using the QuEChERS–HPLC–FLD/DAD Method

The sample preparation and the qualitative and quantitative PAH analysis followed the methodology described by Ciecierska et al. [19], although some modifications were added. These involved PAH extraction and sample clean-up using the QuEChERS method and chromatographic analysis with the HPLC–FLD/DAD technique. The QuEChERS method, as a multi-residue method, allows for the determination of contaminants belonging to different chemical groups simultaneously, and compared to alternative and classical methods of extraction and sample purification, it does not provide adequate sample clean-up from interferences for every food matrix. Therefore, its various modifications were used.

To ensure representativeness and increase the contact area with the reagent, grilled chicken breast tenderloin samples were homogenized using an IKA A11 basic laboratory grinder (IKA Poland Sp. z o. o., Warsaw, Poland) before the extraction and clean-up process.

For fat and PAH extraction, 5 g of homogenized sample was weighed into a Falcon centrifuge tube (50 mL), 10 mL of acetonitrile was added, and the mixture was shaken on a vortex for 1 min. Next, 4 g of magnesium sulfate and 1 g of sodium chloride were added to the tube, vortexed for 3 min, and centrifuged for 3 min at 3400 rpm (MPW—352R laboratory centrifuge, Warsaw, Poland). Subsequently, 4 mL of the upper phase was transferred into a centrifuge tube (15 mL) containing clean-up sorbents: 900 mg of MgSO_4_, 300 mg of PSA, 150 mg of C18, and 10–20 mg of GCB (depending on the color of the extract). The Falcon tube’s contents were intensively mixed using a vortex (3 min) and subsequently centrifuged at 3400 rpm (3 min). The supernatant was subjected to filtration through a PTFE filter (0.2 µm pore diameter) into a chromatography vial and then injected into a chromatographic column in the HPLC–FLD/DAD system.

The analyses of 4 light and 15 heavy PAHs in grilled chicken breast tenderloin samples were performed according to the method outlined by Ciecierska et al. [19], with the use of a Nexera Shimadzu HPLC (LC–40DXR, Kyoto, Japan) with an RF–20 XL fluorescence detector and an SPD–M10AVP diode-array detector. Data were collected and analyzed with the LabSolution 2.1 program. Chromatographic separations were conducted at 30 °C on a Kinetex–PAH column (150 mm × 4.6 mm × 3.5 µm, Phenomenex, Warsaw, Poland) under gradient conditions, with acetonitrile (A) and water (B) (70:30, *v*/*v*), at a flow rate of 1.5 mL/min. The following program of gradient elution was used: 0–3 min 70% A to 73% A, 3–10 min 73% A to 100% A, 10–15.5 min 100% A, 15.5–16.5 min 100% A to 70% A, 16.5–23.0 min 70% A/30% B.

Different fluorescence detection parameters were used. The excitations and emission wavelengths (Ex/Em) for PAH detection were as follows: 256/370 (Anthr, Phen), 270/420 nm (F, Pyr, B[a]A, Chr, 5–MChr, B[b]F, B[k]F, B[a]P, D[ah]A, D[al]P, B[ghi]P, and D[ae]P), 270/470 nm (D[ai]P and D[ah]P), and 270/500 nm (B[j]F and I[cd]P). The diode-array detector (DAD), with a wavelength of 254 nm, was used to detect C[cd]P.

### 2.7. Quantification and Validation of the QuEChERS–HPLC–FLD/DAD Method

Qualitative and quantitative PAH analysis, utilizing an external standard method and validating the QuEChERS–HPLC–FLD/DAD method, was performed following the procedure previously described by Ciecierska et al. [19]. Two PAH standard mixtures (PAH-Mix 183 and PAH-Mix 9, Dr. Ehrenstorfer) were used. Six standard solutions of PAHs at varying concentrations (0.5–50.0 μg/L) were examined to establish the calibration curves for individual PAHs. High correlation coefficients proved the method’s linearity for almost all PAHs in the analyzed concentration range (Table 3). Validation parameters for the 19 analyzed PAHs, such as the limit of detection (LOD), the limit of quantification (LOQ), recovery values along with the relative standard deviation (RSD), and HORRAT_R_ values, were determined according to Commission Regulation (EU) No. 836/2011 [21]. To perform recovery experiments, samples of the grilled chicken tenderloins (unmarinated and grilled on a ceramic contact grill) were fortified with PAHs at three concentration levels of standard mixtures (1, 10, and 100 μg/kg). The fortified and unfortified samples were examined in triplicate. The QuEChERS-HPLC–FLD/DAD method’s performance for PAH analysis in grilled chicken tenderloin samples is presented in Table 3.

All validation parameters of the applied QuEChERS–HPLC–FLD/DAD method, such as LOD, LOQ, recovery, and HORRAT_R_, demonstrated that it complied with the requirements of Commission Regulation (EU) No. 836/2011 [21] for the analysis of 4 marker PAHs in food. Additionally, satisfactory parameters of the method’s performance were achieved for the analyzed polyarenes both from the SCF list and the 4 light EPA PAHs, as shown in Table 3. Figure 1 depicts the chromatograms of the analyzed heavy PAHs from chicken breast tenderloin samples marinated in different marinades and unmarinated, grilled on the charcoal grill without a tray.

### 2.8. Statistical Analysis

The results were statistically analyzed using Statistica ver. 10 PL (StatSoft, Inc., Tulsa, OK, USA). To determine whether or not the data followed a normal distribution, a comparison was made if the *p*-value was ≤0.05. The null hypothesis was rejected, and to estimate the significance of differences in the mean PAH contamination levels among different analyzed variants of grilled products, a two-way analysis of variance and Tukey’s test at a significance level of α = 0.05 were applied. Pearson correlation analysis was also utilized for the weight loss and the color components, as well as the contents of 15 heavy PAHs and 19 PAHs.

## 3. Results and Discussion

### 3.1. Analysis of the Meat’s Weight Loss After the Grilling Process

Due to the grilling process, the weight of the analyzed products decreased (Table 4). Cooking loss was determined within a range of 15.7–39.0%, depending on the type of the analyzed sample. Grilled chicken breast tenderloins marinated in universal, chicken, and honey mustard marinades significantly showed the highest cooking loss on the charcoal grill without a tray (25.7–39.0%). For samples without a marinade, the highest cooking loss was found on two grilling tools: the cast-iron electric contact grill (32.2%) and the charcoal grill without a tray (32.8%). Moreover, comparing only charcoal grills, the results showed that despite sample variants, weight loss was significantly higher for charcoal grills without a tray. Furthermore, comparing the two types of electric grills showed that the weight losses for samples prepared on the cast-iron electric contact grill were significantly higher (20.0–33.6%) than on the ceramic grill (16.2–27.5%).

According to the Pearson correlation analysis, as the weight loss increased, the contamination with the 15 heavy PAHs showed a significant, strong positive correlation (*p* < 0.001). However, comparing the results obtained for the correlation of weight loss and the sum of the 19 PAHs, no correlation was observed (*p* = 0.117), which may have been due to greater variation in the contents of light PAHs in the samples.

The differences in the weight loss of chicken breast tenderloin samples prepared in various marinades mainly depended on the grilling temperature and the cooking method. For direct heating, the charcoal grill without a tray was used. These samples were cooked faster than chicken prepared on a charcoal grill with an aluminum tray. The indirect heating method limited the contact of the meat with the flames, leading to significantly lower weight loss than the direct cooking method. Furthermore, compared to charcoal grilling tools, using the electric grills showed that preparing the meat on the cast-iron contact grill caused significantly higher weight loss than cooking on the electric ceramic contact grill. This result could be explained by the fact that the iron grill’s corrugated surface allowed for greater evaporation of water from the product. Moreover, higher weight loss was also caused by the pressure force of the heating surface on the product and the higher temperature of the cast-iron grill (200–220 °C) than the ceramic grill (180–200 °C).

The results were consistent with those of Ormian et al. [22], whose studies on chicken breast showed significantly higher weight loss in samples prepared at 85 °C (31.4%) than in samples grilled at 75 °C (24.7%). Similarly, Suleman et al. [23] confirmed that the choice of cooking method significantly impacts meat’s weight loss. Lamb patties on a charcoal grill had a significantly higher weight loss than samples grilled in an electric oven or cooked with superheated steam. Furthermore, the obtained results are also consistent with the conclusions formulated by Purslow et al. [24]: that rising temperatures cause weight loss during cooking.

### 3.2. Color Analysis

The effects of different marination treatments and grilling tools on the color of chicken breast tenderloins are shown in Table 5. Our results indicated that the different marinades and cooking methods affected the samples’ color parameters. Examining the L* parameter, it was observed that for the charcoal grill without a tray, samples marinated in honey mustard, universal marinade, and without treatment were significantly darker (38.74, 32.88, and 57.43, respectively) than on other types of appliances. Because of the higher temperature and direct contact with the flames on this grill, most samples showed a significantly darker color on the surface after the grilling process. The redness value (a*) showed that unmarinated samples grilled on a charcoal grill with a tray (3.79) and on a ceramic grill (3.85) had the lowest values of this parameter. Using a charcoal grill without a tray showed significantly lower redness values for the samples without marination (11.55) and those marinated in the universal marinade (9.87). The cast-iron contact grill results showed a lower a* value for three types of samples: universal, chicken, and without marination (7.87, 9.93, and 5.27, respectively). The differences in redness may have been due to the grill design, including corrugated surfaces, the grilling temperature, and the uneven color of the marinades used. Moreover, the results corresponded to the conclusion formulated by Suman et al. [25]: that during grilling, temperatures above 70 °C cause complete denaturation of myoglobin, so the value of the a* parameter decreases and the b* parameter increases.

To compare the color changes before and after grilling, the ΔE parameter was determined (Table 6). For each sample, it was noted that even the inexperienced observer could notice the color differences (ΔE > 5).

The Pearson correlation of color components and PAHs in chicken breast tenderloins showed a weak, negative correlation between the L* parameter and the 19 PAHs (*p* = 0.043) and a weak, positive correlation between a* and the 15 heavy PAHs (*p* = 0.011) as well as the 19 PAHs (*p* = 0.040). There was no significant correlation between lightness and the sum of the 15 heavy PAHs, or between yellowness and the sum of the 15 heavy and 19 overall PAHs.

According to the results obtained in this work, the type of grill and the marination treatment significantly affected the color parameters of the analyzed samples. Moreover, Pearson correlation leads to the conclusion that the lower the proportion of redness (a*), the higher the PAH content in the sample, and the lighter the sample, the lower the contaminations of light and heavy PAHs. The results of this work are consistent with the findings reported by many authors. Silva et al. [26] reported that after the grilling process, the meat becomes darker, and the proportion of redness and yellowness increases. Wongmaneepratip et al. [27], in their studies on chicken breast grilled on a charcoal grill, confirmed that different marinades significantly affected the color parameters. Furthermore, according to Ormian et al. [22], the higher temperature in the geometric center substantially affects the color parameters, lowering the L* and a* parameters and increasing the b* parameter.

### 3.3. Analysis of PAH Contamination in Grilled Meat Samples

#### 3.3.1. Effects of Various Grilling Tools on PAH Formation in Chicken Breast Tenderloins

Results of the contents of PAHs in chicken breast tenderloin samples prepared on four types of grilling tools are presented in Figure 2. Data for the sum of all analyzed PAHs, including the sum of 15 heavy SCF PAHs, 4 heavy/marker PAHs, and B[a]P, are given. The obtained results confirmed the statistically significant differences in the contents of analyzed PAHs between each type of grilling tool. Significantly, the highest levels of a total of 15 PAHs, 4 marker/heavy PAHs, and B[a]P were observed in the products prepared on a charcoal grill without a tray (6.31–8.84 µg/kg, 3.59–5.17 µg/kg, and 1.06–1.52 µg/kg, respectively). Modification of the charcoal grill by using an aluminum tray significantly decreased the levels of these PAHs (3.31–5.21 µg/kg, 1.53–2.68 µg/kg, and 0.51–0.93 µg/kg, respectively). Considering samples prepared on two types of charcoal grill, it can be concluded that direct contact of the samples with the open flames is the main factor that promotes PAH formation. Using an aluminum tray protected the meat samples from direct contact with the fire, leading to significantly lower PAH contamination. Furthermore, the aluminum tray on the charcoal grill prevented the melting fat from the chicken from dropping onto the hot coals. During such a thermal process, chemical modification of organic compounds like lipids generates toxic compounds via pyrolysis, including PAHs, which are carried by smoke and accumulate on the surface of the processed food [1].

Although the variation in PAH levels detected in heat-processed foods is determined by many factors, including the temperature and time of the thermal process, type of fuel used, and proximity or direct contact with the heat source, the contents of those carcinogenic compounds may differ due to the type and fat content of the food. Electric grilling can be an alternative grilling method that significantly decreases levels of PAHs. In the current study, for comparison to charcoal grills, chicken breast tenderloins were also prepared on cast-iron and ceramic electric grills. The contamination levels of the chicken breast tenderloins were significantly higher for the electric cast-iron contact grill (15 heavy PAHs: 2.87–4.12 µg/kg, 4 marker PAHs: 1.67–2.16 µg/kg, B[a]P: 0.51–0.75 µg/kg) than for the electric ceramic contact grill (15 heavy PAHs: 0.69–3.55 µg/kg, 4 marker PAHs: 0.31–1.89 µg/kg, B[a]P: 0.05–0.65 µg/kg). The concentrations of all analyzed heavy PAHs in meat marinated in the universal marinade and grilled on the electric ceramic contact grill (Figure 2a) were below the method’s detection limit. The fact that could explain this result is that the cast-iron contact grill had a corrugated surface, and the fat accumulated in grooves and recesses promoted the concentrations of PAHs. The ceramic contact grill had a smooth bottom surface and a lower grilling temperature. Consequently, the fat melted less. Moreover, the melting fat did not accumulate on its surface.

The concentration levels of B[a]P and the four heavy/marker PAHs in the analyzed samples did not exceed the requirements of Commission Regulation (EU) No. 915/2023 of 25 April 2023 for grilled meat products (5 µg/kg for B[a]P, 30 µg/kg for the four heavy PAHs) [20]. Therefore, it can be concluded that each final product did not pose a health problem.

According to the current study’s results and other authors’ conclusions, the highest PAH concentrations in charcoal grill samples may result from direct heating and the high temperature of the coals (reaching 900 °C) [2]. Modification of the charcoal grill by using an aluminum tray showed a significantly lower concentration of PAHs, similar to those found using an electric cast-iron contact grill or an electric ceramic contact grill. Different studies have confirmed that the method of grilling significantly affects the PAH contamination in cooked products. Most studies, however, as previously mentioned, focused on the determination of PAHs from the US EPA list. Badyda et al. [28], in their study, noted that the concentration of the 15 EPA PAHs was considerably higher in meat grilled over charcoal briquettes (48.19 µg/kg) and ten times lower on the electric grill (4.25 µg/kg). A similar study was conducted by Fatma et al. [29] on grilled broiler chicken. The highest mean values of benzo[a]pyrene, the sum of four PAHs, and total PAHs, mostly from the EPA list, were detected in charcoal grill samples (1.38 µg/kg, 3.09 µg/kg, and 36.0 µg/kg, respectively), while preparing meat on the electric grill generated the lowest average levels of these PAHs (0 µg/kg, 0.44 µg/kg, and 26.4 µg/kg, respectively). These results are consistent with those of Hamzawy et al. [30], whose study showed an almost 100% reduction in the mean concentration of B[a]P in electric grilled chicken and Σ 16 EPA PAHs at the level of 0.04 µg/kg, as opposed to traditional charcoal-grilled samples, for which the Σ 16 EPA PAHs was equal to 1.60 µg/kg. These results correspond to the conclusions formulated by Onwukeme et al. [31], who found that the levels of PAHs are strongly affected by the cooking method. The barbecued and fried samples were the most contaminated by the 16 EPA PAHs (0.20 µg/kg and 10.84 µg/kg, respectively), compared to boiled and roasted chicken (0.14 µg/kg and 0.18 µg/kg, respectively). Furthermore, Husseini et al. [32], in their study, noted that grilled thighs, which have a larger amount of lipids, contained higher amounts of four heavy PAHs (2.89–49.90 µg/kg) than grilled chicken breast (1.52–42.74 µg/kg).

#### 3.3.2. Effects of Various Marinades on PAH Formation in Grilled Chicken Breast Tenderloins

In this study, three different marinades were used to reduce the levels of PAHs in grilled chicken breast tenderloins. The results of the mean contents of PAHs in meat products prepared with the three different marinades, along with unmarinated samples, are presented in Figure 3. Data on the sum of analyzed PAHs, including 15 SCF PAHs, 4 heavy/marker SCF PAHs, and B[a]P, are given.

The obtained results confirmed the statistically significant differences in the contents of PAHs between marinated and unmarinated samples. The results for the charcoal grill without a tray (Figure 3a) illustrated that the highest total concentration of the 15 PAHs was observed in chicken breast tenderloins marinated in the honey mustard marinade (8.84 µg/kg), while the lowest was observed in the chicken marinade (6.31 µg/kg). Similar to a charcoal grill without a tray, samples grilled using an aluminum tray (Figure 3b) showed the highest concentration of PAHs in the honey mustard marinade (5.29 µg/kg). However, the lowest concentration was observed in the universal marinade (3.31 µg/kg). According to the concentrations of PAH4 and benzo[a]pyrene, the highest contamination levels of these compounds were observed in the honey mustard marinade samples (5.02 µg/kg and 1.52 µg/kg, respectively) and unmarinated samples (5.17 µg/kg and 1.28 µg.kg, respectively). Significantly, the lowest contamination levels of the sum of four heavy PAHs and B[a]P were observed in the universal (3.71 µg/kg and 1.06 µg/kg, respectively) and chicken marinades (3.39 µg/kg and 1.08 µg/kg, respectively). However, after using a tray, contamination in the honey mustard marinade was significantly higher (2.68 µg/kg and 0.93 µg/kg, respectively) than in samples without marination treatment (2.21 µg/kg and 0.76 µg/kg, respectively). The contamination levels in samples prepared on an electric ceramic grill (Figure 3c) for the sum of 15 PAHs and 4 heavy PAHs were significantly different for each variant. Similar to the charcoal grill without an aluminum tray, the highest PAH contamination was observed in the honey mustard marinade (2.91 µg/kg and 1.57 µg/kg, respectively), and the lowest was observed in the chicken marinade treatment (0.69 µg/kg and 0.31 µg/kg, respectively). Comparing these samples regarding B[a]P concentration, the chicken marinade showed the lowest concentration of this compound (0.05 µg/kg). However, for B[a]P, there was no significant difference between the honey mustard marinade and the sample without treatment (0.55 and 0.65 µg/kg, respectively). For the universal marinade, heavy PAHs were not detected at all in meat prepared on an electric ceramic grill. Regarding contamination with the 15 total and 4 heavy/marker PAHs in the chicken breast tenderloin samples grilled on a cast-iron contact grill (Figure 3d), it was noted that the universal and chicken marinades led to significantly lower levels of contamination (2.87–3.19 µg/kg and 1.67–1.76 µg/kg, respectively). In samples without marination treatment or grilled in the honey mustard marinade, the concentrations were significantly higher (4.04–4.12 µg/kg and 2.05–2.16 µg/kg, respectively) for the sum of 15 PAHs and 4 marker PAHs. However, for each variant of the samples, there were no significant differences in the concentration of B[a]P (0.51–0.75 µg/kg).

Considering applicable legal requirements, the concentrations of B[a]P and four heavy PAHs in all samples were within the limits established by Commission Regulation (EU) No. 915/2023 [20].

Although none of the tested samples exceeded the limits, based on our research and previous studies, proper preparation of meat and grilling equipment can reduce human exposure to PAHs, especially heavy PAHs. Reducing exposure to polycyclic aromatic hydrocarbons, including food intake, can reduce short-term health problems such as inflammation, nausea, and vomiting, as well as long-term effects such as skin and lung cancer, DNA and liver damage, or negative effects on the circulatory system [1,5,9].

Other studies have also reported that marinating food can significantly decrease the concentrations of PAHs. In a study conducted by Wang et al. [33], it was found that chicken wings that were marinated in six different beer marinades showed significantly lower concentrations of B[a]P (0.30–1.17 µg/kg) and PAH8 (4.31–8.89 µg/kg) than unmarinated samples (B[a]P: 2.05 µg/kg; PAH8: 13.03 µg/kg). The same author’s research [34] on chicken wings marinated in various phenolic marinades with different concentrations of gallic, ferulic, and protocatechuic acids confirmed that marinating samples by adding 0.1–5 mg/L of acids before the grilling process can effectively inhibit the formation of B[a]P and PAH8 (B[a]P: 2.08–2.95; PAH8: 7.66–10.66 µg/kg). Samples without the addition of any phenolic acid showed significantly higher concentrations of PAHs (B[a]P: 3.27 µg/kg, PAH8: 12.83 µg/kg). The obtained results corresponded to the conclusions formulated by Yunita et al. [35], to the effect that a lemon marinade can reduce PAH content. Chicken satay without a lemon marinade showed the highest concentrations of B[a]P and PAH4 (5.61 µg/kg and 14.96 µg/kg, respectively). Regarding samples with different percentages of the lemon marinade, the sample with the highest lemon concentration (15%) showed the lowest levels of carcinogens (B[a]P: 3.72 µg/kg; PAH4: 8.79 µg/kg). Farhadian et al. [36] explored the effects of seven marinades on the formation of B[a]P, B[b]F, and F in grilled beef meat. The meat samples treated with an acidic marinade (with 1.2% lemon juice) achieved a 70% reduction in total PAH concentration. Cordeiro et al. [37] showed that elderberry and white wine vinegar significantly reduced total contamination with 4 PAHs in pork grilled on a traditional charcoal grill. In the unmarinated sample, the sum of 4 indicator PAHs exceeded the legal limit, reaching 31.47 μg/kg. Wongmaneepratip et al. [27] conducted research on pork grilled on a traditional charcoal grill, previously marinated with the addition of two antioxidants: diallyl disulfide (DADS) and quercetin. A significant effect of adding both antioxidants on reducing PAH contamination was demonstrated. Furthermore, Min et al. [38], in a meat model system, also noted that different antioxidants, including BHT, BHA, and α-tocopherol, led to a 37.46–49.43% reduction in the contents of 8 PAHs in different antioxidant-treated samples compared to controls.

The results obtained in this work and other studies confirm that marinating chicken meat with spices and other food additives rich in phenolic compounds and antioxidants can reduce the levels of polycyclic aromatic hydrocarbons. However, this study has shown that the presence of sugars in marinades can also increase the contamination with PAHs. In most cases, chicken breast tenderloins marinated in the honey mustard marinade showed significantly higher PAH contamination than unmarinated samples. Kazazic et al. [39], in their study monitoring honey’s contamination with PAHs, reported that honey from apiaries located close to industrial areas had higher levels of PAHs (12.58 µg/kg) than honey from hives located in unpolluted areas (total PAHs: 2.68–4.76 µg/kg). PAH concentration levels in honey can be diverse and depend on other factors, including vehicle traffic intensity. Nor Hasyimah et al. [40] noted that samples of grilled beef satay marinated in honey and spices had significantly higher levels of PAH contamination (B[a]P: 4.49–32.60 µg/kg; PAH8: 78.47–164.40 µg/kg) than control samples without a marinade (B[a]P: 2.67–4.60 µg/kg; PAH8: 34.59–89.52 µg/kg). Moreover, samples marinated with *Trigona* sp. honey and spices and grilled at 250 °C and 350 °C showed higher levels of the sum of 15 PAHs from the US EPA list (197.05 and 350.38 µg/kg, respectively) than samples marinated with *Apis mellifera* honey (168.04 and 296.03 µg/kg). The amount of sugars in honey also determines the differences in the concentrations of PAHs between those samples. Raw samples marinated in honey from *Trigona* sp. bees had significantly higher amounts of reducing sugars (2.42 ± 0.11 g/100 g) than samples marinated in honey from *Apis mellifera* bees (1.54 ± 0.04 g/100 g). The researchers explained that higher levels of PAH contamination are caused by changes associated with the Maillard reaction, and that carbohydrates can act as precursors of PAH formation. The Maillard reaction was stated to favor aromatization, dehydrocyclization, caramelization, and fat/sugar degradation, promoting PAH formation in food products. Moreover, temperatures above 110 °C cause thermal degradation of honey components such as glucose and fructose, and chloropropanols are formed, which are precursors to PAH formation [41].

#### 3.3.3. Analysis of PAH Qualitative Profiles

Describing the samples prepared on the charcoal grill with and without an aluminum tray (Figure 4, Appendix A), the four light PAHs constituted 84–94% of all PAHs under investigation, and the 15 heavy PAHs averaged 6–16% of the total contents of the 19 PAHs. The results from this study revealed that the concentrations of PAHs in grilled chicken breast tenderloin samples depend on the marinade used, and that marinades with a high sugar content cause a higher level of contamination with heavy PAHs. Therefore, the highest percentages of the 15 heavy PAHs in the total 19 PAHs’ contents were observed for samples marinated in the honey mustard marinade and averaged 15 and 16% for the charcoal grill with and without an aluminum tray, respectively.

The qualitative profiles of PAH contamination with the 4 light and 15 heavy polyarenes were also measured for two electric grills: the cast-iron electric grill with a corrugated top and bottom surface, and the ceramic electric grill with a smooth bottom surface (Figure 5, Appendix A).

The analysis of the tested PAHs on the cast-iron contact grill revealed that the 15 heavy PAHs constituted 18 to 27% and the 4 light PAHs averaged 73 to 82%. Chicken breast tenderloin samples marinated in the honey mustard marinade and without any marinade showed higher contamination with heavy PAHs (27%) than the other two samples prepared in the universal and chicken marination treatments (19% and 18%, respectively). The analysis of the electric ceramic grill showed differentiation in the heavy and light hydrocarbon percentages between the samples. As with the cast-iron grill samples, the qualitative profiles of chicken breast tenderloins marinated in the honey mustard marinade and without marination showed a higher share of the sum of 15 heavy PAHs (24% and 23%), while a lower share was observed in the chicken marinade (5%). Moreover, in the universal-marinated samples, only light PAHs were detected. The cast-iron grill applied in this work caused higher percentages of the 15 heavy SCF PAHs in each of the samples, which may be explained by the high pressure of the grilling plates on the product and the corrugated bottom surface of the appliance, which caused the accumulation and pyrolysis of fat. Therefore, the abovementioned results and studies conducted by other authors confirm that the PAH contamination profile is influenced by many factors [40], including the type of grill, design of the grilling tools, grilling conditions, and the marinade treatment used.

## 4. Conclusions

In conclusion, this study shows that chicken poultry meat subjected to heat treatment on a grill can be a source of carcinogenic compounds such as PAHs. Conscious selection of appropriate equipment for grilling, e.g., electric grills instead of charcoal grills, and marinating poultry meat before cooking, e.g., in universal or chicken marinades rich in phenolic compounds, are extremely important for consumers and food producers, allowing for the minimization of the harmful impacts of PAHs on human health. Charcoal grills, due to direct contact of the chicken breast tenderloins with the heat source, cause the highest levels of PAH contents in processed meat. Modification of charcoal grills by using an aluminum tray or indirect grilling methods and the selection of suitable processing parameters can reduce PAH contamination. In addition, key strategies for consumers to minimize PAHs in grilled poultry meat involve using appropriate marinades before grilling, especially those rich in phenolic compounds. Therefore, considering the high consumption rate of grilled meat, especially in the summer season, by following the rules of safe grilling and the abovementioned methods for PAH minimization, consumers can reduce the amounts of PAHs and ensure the safety of consumed products. Further research is needed to extend the results obtained here regarding minimizing PAH contamination levels in industrial meat processing, or to investigate the effects of other types of marinades and grilling methods. Moreover, additional scientific research on strategies for reducing polycyclic aromatic hydrocarbon contents in various food products will be essential to enhance food safety, protect public health, and minimize PAH contents in food following the ALARA principle, as low as reasonably achievable.

## Figures and Tables

**Figure 1 foods-13-03378-f001:**
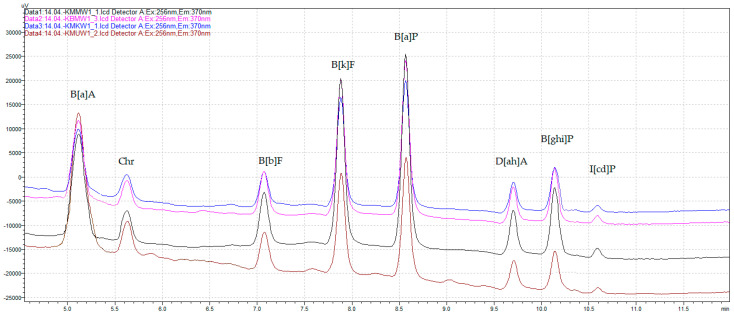
The HPLC–FLD chromatograms of heavy SCF PAHs from chicken breast tenderloin samples marinated in universal marinade (brown), honey mustard marinade (black), chicken marinade (blue), and without marination (pink), grilled on the charcoal grill without a tray.

**Figure 2 foods-13-03378-f002:**
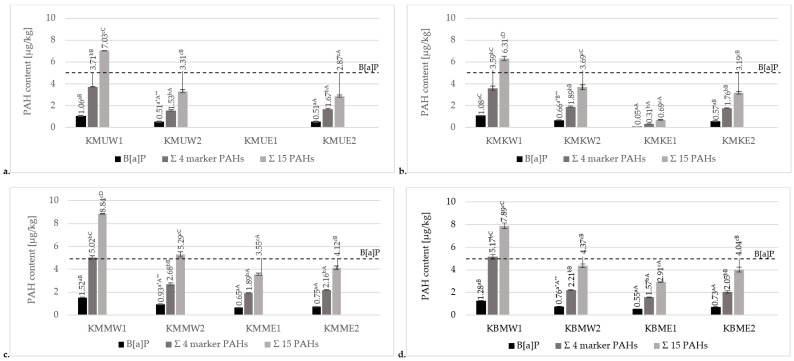
Comparison of mean contents of B[a]P, 4 marker SFC PAHs, and 15 total PAHs for chicken breast tenderloin samples grilled on the charcoal grill without a tray (W1), charcoal grill with an aluminum tray (W2), ceramic contact grill (E1), and cast-iron grill (E2), marinated in various marinades: (**a**) universal marinade (KMU), (**b**) chicken marinade (KMK), (**c**) honey mustard marinade (KMM), and (**d**) without marination (KBM). * Different values for various marination treatments used in the same method of grilling followed by different lowercase letters (a–c) are significantly different at the α = 0.05 level. ** Different values for various grills in the same marination treatment followed by different capital letters (A–D) are significantly different at the α = 0.05 level.

**Figure 3 foods-13-03378-f003:**
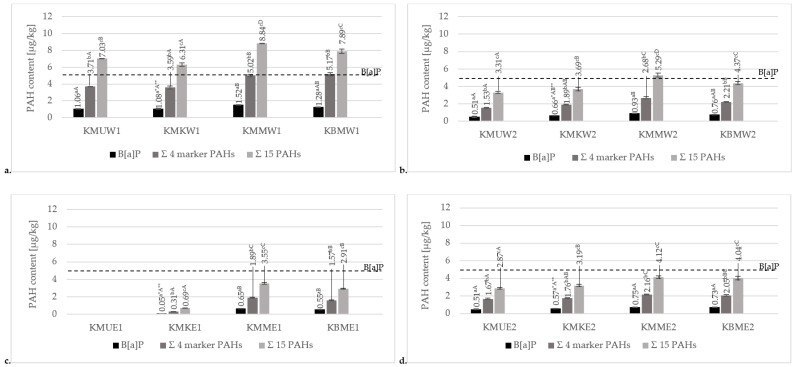
Comparison of mean contents of B[a]P, 4 marker SFC PAHs, and total 15 PAHs for chicken breast tenderloin samples marinated in universal, chicken, and honey mustard marinades or without marination (KMU, KMK, KMM, and KBM, respectively), grilled on (**a**) a charcoal grill without a tray (W1), (**b**) a charcoal grill with an aluminum tray (W2), (**c**) a ceramic contact grill (E1), or (**d**) a cast-iron grill (E2). * Different values for various grills in the same method of marination treatment followed by different lowercase letters (a–c) are significantly different at the α = 0.05 level. ** Different values for various marination treatments in the same method of grilling followed by different capital letters (A–D) are significantly different at the α = 0.05 level.

**Figure 4 foods-13-03378-f004:**
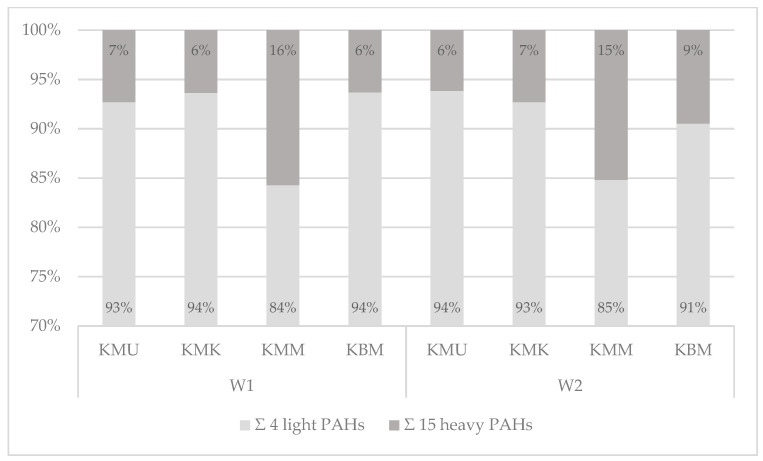
The qualitative profiles of the 4 light and 15 heavy PAHs’ contents in the 19 PAHs in chicken breast tenderloins prepared on the charcoal grill without a tray (W1) and the charcoal grill with an aluminum tray (W2).

**Figure 5 foods-13-03378-f005:**
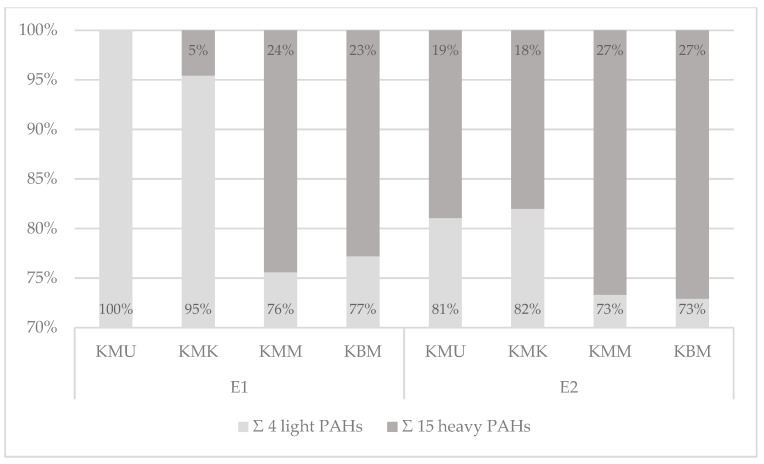
The qualitative profiles of the 4 light and the 15 heavy PAHs’ contents in the 19 PAHs in chicken breast tenderloins prepared on the electric ceramic contact grill (E1) and electric cast-iron contact grill (E2).

**Table 1 foods-13-03378-t001:** Composition of marinades.

Marinade Treatment	Ingredients [g/100 g of Meat]	Code
Universal marinade	5 g universal seasoning (composition: salt, rosemary (10.3%), basil, sugar, onion, paprika, oregano, marjoram, ground mustard, coriander, sunflower oil, thyme, lemon juice, turmeric, cayenne pepper), 5 g refined rapeseed oil	MU
Chicken marinade	5 g chicken seasoning (composition: salt, coriander (11.0%), granulated garlic, sweet pepper (6.5%), rosemary, fenugreek (3.0%), basil (2.2%), black pepper, chili, turmeric, ginger, thyme), 5 g refined rapeseed oil	MK
Honey mustard marinade	5 g spicy mustard (water, white mustard, spirit vinegar, black mustard, sugar, salt, aroma, turmeric extract), 3.5 g lime honey, 1 g refined rapeseed oil, 0.5 g freshly squeezed lemon juice	MM

**Table 2 foods-13-03378-t002:** Grilling parameters.

Type of Grill	Grill Temperature [°C]	Grilling Time [min]	Temperature at the End of the Grilling Process in the Geometric Center of the Product [°C]
W1	240–300	8	88.1
W2	170–220	13	81.3
E1	180–200	10	85.1
E2	200–220	10	87.8

W1—charcoal grill without a tray, W2—charcoal grill with an aluminum tray, E1—electric ceramic contact grill, E2—electric cast-iron contact grill.

**Table 3 foods-13-03378-t003:** The QuEChERS-HPLC–FLD/DAD method’s performance for PAH analysis in grilled chicken breast tenderloin samples.

PAH	Calibration Curve	Correlation Coefficient r^2^	Linearity Range (µg/L)	LOD (µg/kg)	LOQ (µg/kg)	Recovery for 100 µg/kg of Sample Fortification	Recovery for 10 µg/kg of Sample Fortification	Recovery for 1 µg/kg of Sample Fortification	Recovery (%) *	RSD (%) *	HORRAT_R_ Value *
Phen	y = 102,211x + 37,486.4	0.9997	1–50	0.08	0.16	81.5	78.8	76.9	79.1	8.5	0.7
Anthr	y = 62,303.9x + 26,899.6	0.9990	1–50	0.09	0.18	81.8	77.9	74.8	78.2	8.6	0.7
F	y = 23,407.8x + 35,387.1	0.9993	1–50	0.10	0.20	88.3	86.7	81.8	85.6	9.4	0.8
Pyr	y = 268,840x + 86,012.2	0.9997	1–50	0.05	0.10	91.1	88.9	86.6	88.9	9.7	0.8
C[cd]P	y = 221,120x + 11,345	0.9993	2–50	0.35	0.70	107.5	108.9	109.9	108.8	8.7	0.7
B[a]A	y = 120,488x + 36,275.5	0.9997	1–50	0.06	0.12	92.2	88.9	84.8	88.6	7.2	0.6
Chr	y = 39,749.7x + 12,690.7	0.9997	1–50	0.07	0.14	87.5	84.0	82.9	84.8	7.7	0.6
5-MChr	y = 97,566x − 6394.4	0.9999	1–50	0.07	0.14	89.8	83.5	80.2	84.5	7.8	0.7
B[j]F	y = 1600.7x + 196.91	0.9997	2–50	0.25	0.50	82.7	81.2	78.9	80.9	7.5	0.6
B[b]F	y = 64,900x + 19,646.3	0.9997	1–50	0.08	0.16	91.3	86.4	80.5	86.1	8.4	0.7
B[k]F	y = 193,555x + 59,017.2	0.9997	1–50	0.07	0.14	92.8	88.6	84.1	88.5	7.6	0.6
B[a]P	y = 122,195x + 65,266.3	0.9998	1–50	0.07	0.14	91.6	89.8	86.3	89.2	7.7	0.6
D[ah]A	y = 45,718.9x + 11,119	0.9999	1–50	0.12	0.24	85.1	81.5	80.1	82.2	8.2	0.7
D[al]P	y = 359.96x − 154.23	0.9995	2–50	0.20	0.40	81.4	77.2	73.8	77.5	8.4	0.7
B[ghi]P	y = 69,415.7x + 18,240.4	0.9999	1–50	0.14	0.28	88.6	85.2	82.7	85.5	8.5	0.7
I[cd]P	y = 11,025.7x + 1670.83	0.9997	1–50	0.21	0.42	83.6	81.4	78.0	81.0	8.6	0.7
D[ae]P	y = 8144.2x + 456.41	0.9997	1–50	0.22	0.44	82.1	76.9	73.1	77.3	9.4	0.8
D[ai]P	y = 226,619x − 96,869	0.9997	1–50	0.11	0.22	82.3	79.2	73.0	78.1	9.1	0.8
D[ah]P	y = 172,110x − 68,308	0.9997	1–50	0.13	0.26	79.0	73.0	71.0	74.4	9.9	0.8

* Mean recovery, RSD, and HORRAT_R_ values of three different levels of sample fortification. For recovery experiments, *n* = 72 (54 fortified samples of the unmarinated chicken tenderloins grilled on a ceramic contact grill, and 18 unfortified samples).

**Table 4 foods-13-03378-t004:** Weight loss in various marinated and unmarinated grilled chicken breast tenderloin samples.

Universal Marinade		Chicken Marinade		Honey Mustard Marinade		Without Marinade	
Sample Code	Weight Loss [%]	Sample Code	Weight Loss [%]	Sample Code	Weight Loss [%]	Sample Code	Weight Loss [%]
KMUW1	36.6 ± 0.27 ^c^*	KMKW1	25.7 ± 0.28 ^c^	KMMW1	39.0 ± 0.28 ^c^	KBMW1	32.8 ± 0.53 ^c^
KMUW2	15.7 ± 0.36 ^a^	KMKW2	20.4 ± 0.15 ^b^	KMMW2	34.0 ± 0.19 ^b^	KBMW2	18.0 ± 0.46 ^a^
KMUE1	16.2 ± 0.04 ^a^	KMKE1	18.0 ± 0.24 ^a^	KMME1	27.5 ± 0.32 ^a^	KBME1	19.8 ± 0.10 ^b^
KMUE2	23.6 ± 0.05 ^b^	KMKE2	20.0 ± 0.14 ^b^	KMME2	33.6 ± 0.19 ^b^	KBME2	32.2 ± 0.06 ^c^

*n* = 6 (six samples of every kind of product were analyzed). * Different values in the same marination treatment followed by different lowercase letters (a–c), meaning one analyzed comparison, are significantly different at the α = 0.05 level. K—chicken breast tenderloins. MU—universal marinade, MK—chicken marinade, MM—honey mustard marinade, BM—without marinade. W1—charcoal grill without a tray, W2—charcoal grill with an aluminum tray, E1—ceramic contact grill, E2—cast-iron contact grill.

**Table 5 foods-13-03378-t005:** Color of marinated and unmarinated chicken breast tenderloins before and after grilling on different grills.

Cooking Method	Sample	L*	a*	b*
Before grilling	KMU	33.27 ± 1.07 ^a^*^AB^**	6.39 ± 0.25 ^bA^	11.10 ± 0.51 ^bA^
	KMK	36.00 ± 1.08 ^aA^	9.93 ± 0.68 ^bA^	13.06 ± 0.82 ^bA^
	KMM	48.12 ± 0.64 ^bAB^	2.26 ± 0.15 ^aA^	4.60 ± 0.38 ^aA^
	KBM	45.46 ± 2.81 ^bA^	4.45 ± 0.27 ^abA^	3.02 ± 0.63 ^aA^
Charcoal grill without a tray	KMUW1	32.88 ± 1.63 ^aA^	9.87 ± 0.69 ^aA^	24.76 ± 0.42 ^bB^
	KMKW1	48.22 ± 0.77 ^bB^	11.60 ± 1.12 ^abB^	22.28 ± 0.99 ^abB^
	KMMW1	38.74 ± 0.18 ^aA^	15.96 ± 0.07 ^bB^	17.33 ± 0.40 ^aB^
	KBMW1	57.43 ± 0.32 ^cB^	11.55 ± 0.35 ^aB^	24.23 ± 1.93 ^bC^
Charcoal grill with an aluminum tray	KMUW2	46.93 ± 0.74 ^aB^	6.13 ± 0.39 ^bA^	25.99 ± 0.71 ^cB^
	KMKW2	53.84 ± 2.06 ^bB^	7.68 ± 0.19 ^bA^	29.34 ± 0.86 ^cC^
	KMMW2	65.46 ± 0.23 ^cC^	4.95 ± 0.62 ^abA^	23.33 ± 1.29 ^bCD^
	KBMW2	74.79 ± 1.64 ^dC^	3.79 ± 0.52 ^aA^	14.87 ± 0.19 ^aB^
Ceramic contact grill	KMUE1	41.22 ± 1.27 ^aBC^	7.96 ± 0.90 ^bA^	21.44 ± 1.18 ^aB^
	KMKE1	48.86 ± 1.41 ^aB^	11.19 ± 0.45 ^cB^	24.16 ± 1.44 ^acBC^
	KMME1	54.22 ± 1.15 ^bB^	13.43 ± 0.66 ^cBC^	28.50 ± 0.72 ^cD^
	KBME1	69.81 ± 1.83 ^cC^	3.85 ± 0.53 ^aA^	19.72 ± 1.19 ^aBC^
Cast-iron contact grill	KMUE2	43.23 ± 1.57 ^aC^	7.87 ± 0.66 ^aA^	25.06 ± 0.70 ^abB^
	KMKE2	50.86 ± 1.08 ^bB^	9.93 ± 0.42 ^aAB^	27.45 ± 0.60 ^bBC^
	KMME2	52.54 ± 1.30 ^abB^	14.19 ± 0.74 ^bC^	17.19 ± 0.46 ^aBC^
	KBME2	68.68 ± 1.48 ^cC^	5.27 ± 0.18 ^aA^	22.21 ± 1.65 ^abC^

*n* = 6 (six samples of every kind of product were analyzed). * Different values in the same method of grilling followed by different lowercase letters (a–d) are significantly different at the α = 0.05 level. ** Different values in the same marination treatment followed by different capital letters (A–D) are significantly different at the α = 0.05 level.

**Table 6 foods-13-03378-t006:** The difference in color of chicken breast tenderloins before and after grilling.

Charcoal Grill Without a Tray		Charcoal Grill with an Aluminum Tray		Ceramic Contact Grill		Cast-Iron Contact Grill	
Sample	ΔE	Sample	ΔE	Sample	ΔE	Sample	ΔE
KMUW1	14.10 ± 0.20 ^a^*^AB^**	KMUW2	20.21 ± 1.69 ^aC^	KMUE1	13.14 ± 0.91 ^aA^	KMUE2	17.21 ± 2.20 ^aBC^
KMKW1	15.40 ± 0.86 ^aA^	KMKW2	24.26 ± 2.53 ^bC^	KMKE1	17.03 ± 0.87 ^aAB^	KMKE2	20.68 ± 0.81 ^aBC^
KMMW1	20.92 ± 0.48 ^bA^	KMMW2	25.67 ± 1.15 ^bB^	KMME1	27.08 ± 0.92 ^bB^	KMME2	17.90 ± 0.80 ^aA^
KBMW1	25.37 ± 2.29 ^cA^	KBMW2	31.64 ± 3.18 ^cB^	KBME1	29.54 ± 2.87 ^bB^	KBME2	30.14 ± 2.66 ^bB^

* Different values in the same method of grilling followed by different lowercase letters (a–c) are significantly different at the α = 0.05 level. ** Different values in the same marination treatment followed by different capital letters (A–C) are significantly different at the α = 0.05 level.

## Data Availability

The original contributions presented in the study are included in the article/Appendix A, further inquiries can be directed to the corresponding author.

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
