# Peer review of "Effects of Different Marinades and Types of Grills on Polycyclic Aromatic Hydrocarbon Content in Grilled Chicken Breast Tenderloins"

_foods, 2024, doi:10.3390/foods13213378_

Round 1
Reviewer 1 Report
Comments and Suggestions for Authors Dear Authors, I have reviewed your manuscript entitled "Effect of Different Marinades and Types of Grilling on the Content of Polycyclic Aromatic Hydrocarbons in Grilled Chicken Breast Fillets". The manuscript is well prepared, and we have only minor suggestions for improvement. The study is based on a robust and validated methodology, and the experimental design is carefully constructed. The results are presented in a clear and organized manner, and the discussion is coherent and well-founded in the context of the findings. My only recommendation is to include the sample size and number of replicates in all tables (both validation and results) to increase the clarity and reproducibility of the study.
Reviewer 2 Report
Comments and Suggestions for Authors
The manuscript investigates the impact of different marinades and grilling methods on the contamination of chicken breast tenderloins by polycyclic aromatic hydrocarbons (PAHs). The manuscript presents a significant contribution to the ongoing research on minimizing PAH contamination in grilled meat. While the effect of grilling methods and marinades on PAH formation has been studied previously, the authors provide a novel perspective by focusing on chicken breast tenderloins—a commonly consumed meat—and by employing commonly used marinades in Poland. Additionally, the detailed comparison between different types of grilling tools, including charcoal and electric grills, offers practical implications for food safety.

Comments on the Quality of English Language
Review the manuscript for grammatical errors and sentence flow to ensure clarity. Proofreading by a native English speaker or proofreading service should be conducted to improve both language and organization quality.
Reviewer 3 Report
Comments and Suggestions for Authors
This paper has investigated the effect of grills and marinades on chicken breast tenderloins. This paper is well written and include relevant information to be taken into account to minimize the PAHs content in grilled meat.
Comments:
- Page 4, Equation to calculate Colour before and after grilling: Please explain the sense of the symbol ^.
- Clarify some acronyms such in page 5 such as PSA or GC. They have not been defined in the text.
- Figure 2 and 3 show the same data: Figure 2 comparing grill options and the secong¡d one comparing marinades. Please delete one of them.
